# Reaction Mechanism of CA_6_, Al_2_O_3_ and CA_6_-Al_2_O_3_ Refractories with Refining Slag

**DOI:** 10.3390/ma15196779

**Published:** 2022-09-30

**Authors:** Jie Liu, Zheng Liu, Jisheng Feng, Bin Li, Junhong Chen, Bo Ren, Yuanping Jia, Shu Yin

**Affiliations:** 1School of Materials Science and Engineering, University of Science and Technology Beijing, Beijing 100083, China; 2Angang Construction Consortium Co., Ltd., Anshan 114001, China; 3ZiBo City LuZhong Refractory Co., Ltd., Zibo 255000, China; 4Institute of Multidisciplinary Research for Advanced Materials, Tohoku University, 2-1-1Katahira, Aoba-ku, Sendai 980-8577, Japan; 5Advanced Institute for Materials Research (WPI-AIMR), Tohoku University, Katahira 2-1-1, Aoba-ku, Sendai 980-8577, Japan

**Keywords:** corrosion resistance, refining slag, refractory, thermodynamic simulation

## Abstract

In this study, to clarify the corrosion mechanism of CA_6_ based refractory by refining slag, the static crucible tests for CA_6_, CA_6_-Al_2_O_3_, and Al_2_O_3_ refractory, were carried out and the detail reaction processes were analyzed from the perspective of thermodynamic simulation and structural evolution. From the results, CaAl_4_O_7_ plays a vital role in the slag corrosion resistance of the three refractories. Regarding CA_6_ refractory, the double pyramid module in CA_6_ crystal structure was destroyed very quickly, leading to the rapid collapse of its structure to form the denser CaAl_4_O_7_ in high amounts. As a result, a reaction layer mainly composed of CaAl_4_O_7_ formed, which effectively inhibited the slag corrosion, so CA_6_ refractory exhibits the most excellent slag corrosion. Meanwhile, the formation of CaAl_4_O_7_ can also avoid CA_6_ particles entering the molten steel to introduce exogenous inclusions. For Al_2_O_3_ refractory, the generation of CaAl_4_O_7_ is much slower than that of CA_6_ and CA_6_-Al_2_O_3_ refractory, and the amount generated is also quite small, resulting in its worst slag corrosion among the three crucibles. Therefore, CA_6_ based refractory has excellent application potential in ladle refining and clean steel smelting.

## 1. Introduction

During the ladle refining process, the refractory is corroded severely due to the continuous reaction of the refining slag [1,2,3]. The slag corrosion will destroy the structure of the refractory and reduce the service life [4,5,6,7]. More importantly, under the continuous scouring effect of molten steel, the dislodged refractory and the corrosion products would enter the molten steel to form exogenous large size inclusions [8,9,10,11,12,13,14,15,16,17], which will have a fatally harmful effect on the purity of the molten steel and the quality of the final products [17,18,19,20]. Therefore, in order to eliminate the introduction of exogenous inclusion and to increase the service life of the refractories, it is absolutely essential to study corrosion behavior between the refractory and refining slag.

At present, the primary materials for the refining ladle are Al_2_O_3_ system refractories, due to their high density and stable high temperature properties. Many researches have been conducted to investigate the corrosion process between Al_2_O_3_ refractories and the refining slag [21,22,23], and many achievements have been obtained. It was found that the slag corrosion degree of Al_2_O_3_ refractories is influenced by factors such as temperature and refining slag composition [24,25,26,27]. Some scholars pointed out that [28,29] a small amount of CaAl_12_O_19_ generated on the surface of Al_2_O_3_ refractories can prevent further corrosion effectively, which is a critical mechanism. After years of development, an understanding of the corrosion resistance of Al_2_O_3_ refractories has been profound, but there are few reports on the introduction and control of inclusions. So far, as Al_2_O_3_ refractories, the problem of introducing inclusions into molten steel is still unavoidable. With the development of cleanliness steel smelting, the effect of refractories on inclusions has attracted more and more attention. In order to prepare refractory materials with excellent corrosion resistance and less introduction of inclusions, in our previous work [30,31,32,33,34] we have synthesized the pure dense CA_6_ (calcium hexaaluminate, CaAl_12_O_19_) base raw material and studied the slag corrosion resistance. After that, some researchers focused on optimizing CA_6_ structure and tried to dope N^3−^ or Zr^4+^ into CA_6_ [35,36], trying to improve performance further and achieve expected results. Besides that, from our recent experiments, the CA_6_-based refractories not only have excellent slag corrosion resistance but can also reduce the size of inclusions and absorb sulfur in steel. Researches on the preparation and structural improvement of CA_6_ refractory are consummate. However, its slag corrosion mechanism is not clear, and the effect on the exogenous inclusion is not confirmed, which cannot provide strong support for promotion and application. Therefore, it is essential and meaningful to study the slag corrosion of CA_6_-based refractory, which has a great significance on refining ladle and even on the whole steelmaking process.

In this study, the reaction mechanism between two CA_6_-based refractory (pure CA_6_ and CA_6_-Al_2_O_3_ composite) and refining slag was investigated and analyzed, and the Al_2_O_3_ refractory commonly used in the refining ladle was also used as a comparison. The microstructure and distribution of elements in the corrosion area were characterized, and the corrosion process was deduced and simulated by the thermodynamic software. The results of this work proved that CA_6_ refractory has a great application prospect in the metallurgical industry due to excellent and distinctive slag corrosion resistance.

## 2. Materials and Methods

### 2.1. Preparation of the Crucibles and Refining Slag

The CA_6_, Al_2_O_3,_ and CA_6_-Al_2_O_3_ crucibles were prepared with CA_6_ powder (purity > 98 wt%, particle size ≤ 74 μm, Shengchuan, Shandong), Al_2_O_3_ powder (purity > 98 wt%, particle size ≤ 74 μm, Shengchuan, Shandong), and the CA_6_ and Al_2_O_3_ powder (mass ratio CA_6_: Al_2_O_3_ = 1:1), respectively. The static crucible method was adopted for the corrosion resistance test. Firstly, the dried powder was pressed under 30 MPa into 80 × 80 × 80 mm^3^ cubes. Secondly, cylindrical holes with a diameter and height of 40 mm were drilled in the cubes as slag holes. Finally, the crucibles were fired at 1650 °C for 3 h. The properties of the crucibles are shown in Table 1. 

The chemical composition of the refining slag is shown in Table 2. The refining slag was prepared using analytically pure CaO (Sinopharm Chemical Reagent, purity > 99 wt%, particle size ≤ 74 μm), Al_2_O_3_ (Sinopharm Chemical Reagent, purity > 99 wt%, particle size ≤ 74 μm), MgO (Sinopharm Chemical Reagent, purity > 99 wt%, particle size ≤ 40 μm) and SiO_2_(Sinopharm Chemical Reagent, purity > 99 wt%, particle size ≤ 74 μm). 

### 2.2. Experimental

The static crucible method was used to investigate the slag corrosion behavior of the three refractories. Put 70 g refining slag into the crucibles and then heat the crucibles to 1600 °C at a heating rate of 5 °C/min in the box furnace (KSL-1700X-M). After holding for 3h, the furnace stopped working and the crucible was taken out when it cooled to room temperature. The crucibles were cut along the center, and the corrosion area was made into a mosaic sample and then polished. 

### 2.3. Characterization

The composition of the refining slag was analyzed by X-ray fluorescence spectrometry (XRF, Shimadzu, Japan). The micro-morphology of the slag-refractory interface was observed by scanning electron microscopy (SEM, FEI Nova nano 450, USA). The elemental distribution at the slag-resistant material interface was analyzed by SEM equipped with Energy Dispersive Spectrometer (EDS, EDAX Team, USA).

### 2.4. Thermodynamic Simulation 

During the corrosion test, only the final result can be determined clearly and the intermediate process cannot be observed directly. Therefore, thermodynamic simulation for the slag corrosion process is necessary.

In this work, Factsage7.0 was used to simulate the corrosion process of three crucibles. The calculation mechanism is shown in Figure 1. The left side is refining slag and the right side is refractory, and the concentration of the refining slag and refractory at the interface area changes in the reverse cross between 0~1. During the calculations, the interface area is considered as a composition of multiple cross sections and the corrosion process of the refractory is simulated by predicting the generation of each section.

The variable <X> in the Equilib module is used to calculate the inverse crossover interdiffusion, based on which to simulate the corrosion process of the refractory from thermodynamic aspect. When X = 0, the refining slag is all the composition of the system. When x = 1, the refractory is all the composition of the system. At the beginning of the reaction, X was defined as 1. With the decrease of X, more and more slag is involved in the reaction. The oxide data included in the calculation are available in the FToxid module. The calculated temperature was set at a constant 1873 k and the pressure was 1atm.

## 3. Results and Discussion

### 3.1. Composition Changes of Refining Slag 

The composition of slag always changed due to the reaction with refractory. Figure 2 shows the variations of the refining slag composition after the corrosion test. As can be seen from Figure 2, the content of the CaO and MgO in the refining slag decreased, and the content of the Al_2_O_3_ increased in all three crucibles. It indicates that the CaO and MgO in the slag reacted with or entered into the refractories, and Al_2_O_3_ in the refractories diffused into the slag during the corrosion process. The content of the SiO_2_ in the slag remained almost unchanged, which is due to the large ionic radius of the silicate ion. The composition of the slag variation shows that the three crucibles were corroded by the refining slag in varying degrees.

### 3.2. Microstructure and Element Distribution 

#### 3.2.1. Corrosion of CA_6_ Crucible

The microstructure results of the CA_6_ crucible after slag corrosion are shown in Figure 3. The left side is the slag layer and the right side belongs to the original brick layer, and the reaction layer is in the middle, as shown in the red dotted area. It can be seen that many pores exist in the original brick layer. The average diameter of the pores ranged from 12 to 180 μm. In general, the higher porosity, the more severe corrosion of the crucible. However, the CA_6_ crucible shows excellent slag resistance. At the slag-crucible interface, the width of the reaction layer is 50~60 μm, which is the thinnest of the three crucibles (Figures 5 and 7). 

The EDS results show that CaAl_4_O_7_ generated at the slag-refractory interface (reaction layer), and most of the components in the reaction layer are CaAl_4_O_7_ (such as the EDS results of area 1). At the experimental temperature, liquid slag penetrated into the refractory through the pores and reacted with refractory to form CaAl_4_O_7_, which is the main reaction during the corrosion test. The corrosion of the refractory by the liquid slag was effectively inhibited due to the high viscosity of the CaAl_4_O_7_. As the time increased, CaAl_4_O_7_ continued to react with CaO in the liquid slag to form CaAl_2_O_4_ in the reaction layer. In addition, a very small amount of slag phase of Al_2_O_3_-CaO-SiO_2_-MgO was found in the reaction layer. It can be clearly found that the microstructure of the reaction layer was denser than that of the original bricklayer. One of the reasons is the gaps and pores in the reaction layer were filled by CaAl_4_O_7_ and CaAl_2_O_4_, which is also an important reason for preventing further corrosion of the refining slag. To reveal the penetration degree of the refining slag, area 2 was selected randomly for EDS analysis. The very limited Mg and Si were detected, indicating that the CaAl_4_O_7_ layer effectively prevents the penetration of the refining slag.

Figure 4 gives the element distribution result from the slag layer to the original brick layer. It can be seen that the amount of Al in the original brick layer is significantly more than that in the refining slag, while the amount of Ca was the opposite. Thus, it can be deduced that the Ca in the slag diffused into the refractory during the reaction, while Al diffused from the refractory to the refining slag. The amount of Mg and Si in the refractory was minimal, which can also be proved by the EDS result of area 2, indicating that only a very limited Mg and Si in the refining slag penetrate into the crucible through the pores or the gaps between grains.

#### 3.2.2. Corrosion of Al_2_O_3_ Crucible

Figure 5 exhibits the microstructure of the Al_2_O_3_ crucible after the static crucible test. As can be seen from Figure 5, from left to right are the slag layer, reaction layer, penetration layer, and original brick layer, respectively. The width of the reaction layer was 300 μm, and the maximum width of the penetration layer was more than 880 μm that was the widest among the three crucibles.

The EDS results show MgO·Al_2_O_3_ generated in the reaction layer and it was formed by the reaction between MgO in the refining slag and Al_2_O_3_ in the refractory, which is the main component in the reaction layer. Some researchers [4,27] pointed out that the presence of MgO·Al_2_O_3_ at the interface has some hindrance to the penetration of slag. Meanwhile, trace amounts of CaAl_4_O_7_ were also detected in the reaction layer, but its generation mechanism is different from the CA_6_ crucible. The Al_2_O_3_ in the refractory and CaO in the slag reacted to form CaAl_12_O_19_ first, and then the CaAl_12_O_19_ continued to react with CaO in the slag to form CaAl_4_O_7_ in the reaction layer. Compared with the CA_6_ crucible, EDS results (area 3) show that the content of Mg in the reaction layer increased obviously and the amount of Ca decreased. It is confirmed again that the main phase was the MgO·Al_2_O_3_ and the content of CaAl_4_O_7_ was limited in the reaction layer. Meanwhile, it can be noticed that the liquid slag phase was found in the reaction layer, and its content is more than that in the CA_6_ crucible. In the penetration layer, except for the Al_2_O_3_ particles, the CaAl_12_O_19_ is also found in this area, proving that the Ca in the refining slag gradually penetrated into the refractory through the pores or cracks and reacted with the refractory. In addition, some of slag phase of Al_2_O_3_-CaO-SiO_2_-MgO was found in the penetration layer, proving that the reaction layer does not have an advantage in preventing slag penetration. The EDS results of area 4 show that the content of Mg and Si in the penetration layer is more than that of CA_6_ and CA_6_-Al_2_O_3_ crucibles, indicating the Al_2_O_3_ crucible has the worst slag corrosion resistance of the three crucibles. 

Figure 6 is the element distribution result of the slag-crucible interface. It can be seen that the content of Al in the crucible was more than that in slag and Ca was detected in the slag at higher amounts. The element of Ca in the refractory mainly came from the penetration of the refining slag because of the absence of CaO in the raw material. A small amount of Mg and Si were found in the refractory, combined with the EDS result of area 4 in the penetration layer, which was most likely caused by the penetration of liquid slag.

#### 3.2.3. Corrosion of CA_6_-Al_2_O_3_ Crucible

Figure 7 shows the SEM image of the CA_6_-Al_2_O_3_ crucible after the slag corrosion. From left to right are the slag layer, reaction layer, penetration layer, and original brick layer, respectively. From Figure 6, the reaction layer was generated between the refining slag and CA_6_-Al_2_O_3_ crucible, and its width was about 200 μm, which was wider than that of the CA_6_ crucible and thinner than that of the Al_2_O_3_ crucible. The microstructure of the reaction layer was denser compared to the original brick layer. In the reaction layer, CaAl_4_O_7_ is found from the EDS results, and it is quite possibly generated by the reaction between CA_6_ in the refractory and CaO in the refining slag. Of course, a tiny part of CaAl_4_O_7_ may also come from the multistep reaction of Al_2_O_3_ in the refractory and CaO in the refining slag. In the CA_6_ crucible, the CaAl_4_O_7_ can inhibit further corrosion of the refining slag. However, the content of CaAl_4_O_7_ in the CA_6_-Al_2_O_3_ crucible was lower than that in the CA_6_ crucible due to the limit of raw materials, which is one of the reasons for the wider reaction layer than that of the CA_6_ crucible. In the reaction and penetration layer, a similar position to that of the CA_6_ and Al_2_O_3_ crucibles was selected for EDS analysis. The results show the content of Ca in the reaction layer is less than that in the CA_6_ crucible but more than in the Al_2_O_3_ crucible, so it can be deduced that the amount of CaAl_4_O_7_ in the reaction layer is between the other two crucibles. In the penetration layer, the amount of Mg and Si was decreased compared with the Al_2_O_3_ crucible, indicating the slag corrosion resistance of the CA_6_-Al_2_O_3_ crucible is better than the Al_2_O_3_ crucible but worse than the CA_6_ crucible.

Figure 8 is the element distribution result of the CA_6_-Al_2_O_3_ crucible. From Figure 8, the content of Al in the crucible was more than that in slag and Ca was detected in the refractory at higher amounts. The Mg and Si were also detected in the crucible and the content was less than that of the Al_2_O_3_ crucibles, which was also confirmed by the EDS result of area 6, proving the slag resistance of the CA_6_-Al_2_O_3_ crucible was worse than the CA_6_ crucible but better than Al_2_O_3_ crucible.

Through the above analysis, it can be found that the CA_6_ crucible shows the best slag corrosion resistance, followed by the CA_6_-Al_2_O_3_ crucible, and the Al_2_O_3_ crucible is the worst. The reaction mechanism of the CA_6_ and Al_2_O_3_ crucible with the refining slag is shown in Figure 9. It can be seen from Figure 9a–c that the CA_6_ particles reacted with the refining slag to form CaAl_4_O_7_ that can fill the pores or the gaps between the particles. Due to the high viscosity of CaAl_4_O_7_, the slag-refractory interface will be denser and has a positive effect on inhibiting the slag corrosion. At the same time, it should be noticed that the probability of CA_6_ particles entering the molten steel is greatly reduced due to the formation reaction layer, so the introduction of exogenous inclusions from the refractory is greatly reduced. In our recent work, it has been proved that the reaction product of the CA_6_ refractory and refining slag can decrease the number and size of the inclusions. For the Al_2_O_3_ crucible, in addition to reacting with the Al_2_O_3_ particles at the interface, the refining slag also reacts with the inside Al_2_O_3_ crucible by penetrating into the refractory through the pores and the gaps between the Al_2_O_3_ particles, as shown in Figure 9d. During the corrosion process, the Al_2_O_3_ particles gradually dissolve or fall off into the refining slag, causing the more serious corrosion of the crucible, which is shown in Figure 9e,f. In addition, the fall-off particles can enter the molten steel and increase the number of inclusions in steel, which has a negative influence on the quality of steel production. Regarding the CA_6_-Al_2_O_3_ crucible, the CaAl_4_O_7_ is generated faster and the content is more due to the CA6 raw materials compared with the Al_2_O_3_ crucible. Therefore, the slag corrosion resistance of the CA_6_-Al_2_O_3_ crucible is better than that of the Al_2_O_3_ crucible.

### 3.3. Thermodynamic Simulation of Corrosion of Crucibles by Refining Slag

The results of the thermodynamic simulation of the corrosion process of the CA_6_ crucible are shown in Figure 10a. The corrosion process can be divided into the following steps according to the variation of X.

From Figure 10a, when 1.0 < X < 0.76, four phases of CaAl_4_O_7_, Ca_2_Mg_2_Al_28_O_46_, CaAl_12_O_19_, and liquid slag were obtained. With the decrease of X, the content of CaAl_4_O_7_, Ca_2_Mg_2_Al_28_O_46_, and liquid slag increased while the content of CaAl_12_O_19_ decreased sharply. When X = 0.76, the content of Ca_2_Mg_2_Al_28_O_46_ reached a maximum of 40.95% and the content of CaAl_12_O_19_ was zero. Ca_2_Mg_2_Al_28_O_46_ and CaAl_4_O_7_ generated by the reaction of Equations (1) and (2):2CaAl_12_O_19_ + 2MgO + 2Al_2_O_3_ = Ca_2_Mg_2_Al_28_O_46_
(1)
CaAl_12_O_19_ + 2CaO = 3CaAl_4_O_7_(2)

When 0.76 < X < 0.7, a new phase of CaMg_2_Al_16_O_27_ appeared and increased with decreasing X while the content of Ca_2_Mg_2_Al_28_O_46_ gradually decreased until the content was zero at X = 0.7, indicating that Ca_2_Mg_2_Al_28_O_46_ gradually transformed into CaMg_2_Al_16_O_27_ and CaAl_4_O_7_ following the Equations (3) and (4) respectively:Ca_2_Mg_2_Al_28_O_46_ = 2CaAl_4_O_7_ + 2MgO + 10Al_2_O_3_(3)
Ca_2_Mg_2_Al_28_O_46_ + 2MgO + 2Al_2_O_3_ = 2CaMg_2_Al_16_O_27_
(4)

When 0.7 < X < 0.6, the spinel phase appeared in the system at X = 0.66 through the reaction of Equation (5). The content of CaMg_2_Al_16_O_27_ decreased gradually, and the content of CaAl_4_O_7_ increased gradually with the decrease of X, which indicated the CaMg_2_Al_16_O_27_ continues to be converted to the CaAl_4_O_7_ by the equation (6). When X = 0.6, the content of CaMg_2_Al_16_O_27_ was zero, while the content of CaAl_4_O_7_ and spinel reached the maximum content of 49.04% and 8.91%, respectively:Al_2_O_3_ + MgO = MgAl_2_O_4_
(5)
CaMg_2_Al_16_O_27_= CaAl_4_O_7_ + 2MgO + 6Al_2_O_3_(6)

When 0.6 < X < 0.47, the content of the CaAl_4_O_7_ decreased sharply and the liquid slag increased rapidly. When X = 0.47, the content of CaAl_4_O_7_ was zero. When 0.47 < X < 0.14, two phases of liquid slag and spinel were in the system, and the content of spinel decreased to zero at X = 0.14. When 0.14 < X < 0, the liquid slag was the only phase in the system. 

The results of the thermodynamic simulation of the Al_2_O_3_ crucible and CA_6_-Al_2_O_3_ crucible are shown in Figure 10b,c, respectively. The trends of the thermodynamic simulation results for the Al_2_O_3_ and CA_6_-Al_2_O_3_ crucibles are roughly similar to the CA_6_ crucible. However, significant differences existed in the beginning stage of the corrosion. For the Al_2_O_3_ crucible, when 1 < X < 0.76, the Al_2_O_3_ converted to CaAl_12_O_19_ gradually by Equation (7), and the Ca_2_Mg_2_Al_28_O_46_ can also be generated through Equation (8). When X = 0.76, the phase of CaAl_4_O_7_ generated in the system and its content reached the maximum of 34.18% when X = 0.48, which is lower than that of the CA_6_ and CA_6_-Al_2_O_3_ crucible. For the CA_6_-Al_2_O_3_ crucible, the Al_2_O_3_ converted to CaAl_12_O_19_ first, and then CaAl_12_O_19_ started to react with the refining slag to form CaAl_4_O_7_ at X = 0.86. And the maximum content of CaAl_4_O_7_ is 40.11% when X = 0.54, which is higher than that of the Al_2_O_3_ crucible but lower than that of the CA_6_ crucible: CaO + 6Al_2_O_3_= CaAl_12_O_19_
(7)
2CaO + 2MgO + 14Al_2_O_3_= Ca_2_Mg_2_Al_28_O_46_(8)

From the analysis of thermodynamic simulation, it can be found that the high melting point phase CaAl_4_O_7_ is the critical point in enhancing the slag resistance of the crucible [37,38,39,40]. The maximum content of the CaAl_4_O_7_ of the CA_6_, Al_2_O_3_, and CA_6_-Al_2_O_3_ crucible is 49.04%, 34.18%, and 40.11%, respectively, through the thermodynamic simulation results, which the trend is consistent with the EDS results. The CaAl_4_O_7_ was formed at the beginning of the reaction for the CA6 crucible. For the Al_2_O_3_ and the CA_6_-Al_2_O_3_ crucible, the CaAl_4_O_7_ was formed at X = 0.54 and 0.86, respectively. Therefore, the CaAl_4_O_7_ generated fastest and the amount was the most during the corrosion process of the CA_6_ crucible, which had a great benefit on the slag resistance corrosion.

### 3.4. Crystal Structure Analysis of the Refractories

The results of the EDS analysis and thermodynamic simulations show that the CA_6_ crucible can rapidly generate a higher amount of CaAl_4_O_7_ by reacting with the refining slag, while the Al_2_O_3_ crucible generates the CaAl_4_O_7_ through a series of reactions. In this section, the crystal structure of the CA_6_ and Al_2_O_3_ is analyzed, and the reason for the difference in the CaAl_4_O_7_ generation rate and quantity between the CA_6_ and Al_2_O_3_ is also explained.

Figure 11 shows the crystal structure of the CA_6_ and Al_2_O_3_. It can be seen that one Al and five O combined to form the double pyramid module in the mirror layer of the CA_6_ crystal structure, as shown in the area circled red dotted line. The double pyramid module is the active site of CA_6_ and is unstable during the reaction process. When the refining slag reacts with the crucible, the active site will be destroyed quickly, resulting in the rapid collapse of the CA_6_ crystal structure to form the denser CaAl_4_O_7_. The crystal structure of CaAl_4_O_7_ is more stable and can effectively inhibit the further corrosion of slag, which is one of the reasons for the excellent slag corrosion resistance of the CA_6_ crucible. Al_2_O_3_ crystal is an octahedral structure and more stable compared to CA_6_, so the reaction rate with Ca in the refining slag is very slow, resulting in a low content of CaAl_4_O_7_ in the reaction layer and poor corrosion resistance. More importantly, Al_2_O_3_ particles with high stability may directly enter the refining slag and molten steel, resulting in the generation of exogenous inclusions, which has a negative impact on the control of inclusions. At this point, the CA_6_ refractory can well avoid this problem. So, the excellent slag corrosion resistance of the CA_6_ crucible has a great application prospect in the ladle refining process, especially for the smelting clean steel.

## 4. Conclusions

Three crucibles (CA_6_ crucible, Al_2_O_3_ crucible, and CA_6_-Al_2_O_3_ crucible) were selected to investigate the corrosion resistance of the refining slag through laboratory experiments and thermodynamic simulations. The following conclusions were obtained.

(1) The three crucibles show different slag corrosion resistance, CA6 crucible has the best corrosion resistance, followed by the CA_6_-Al_2_O_3_ crucible. The Al_2_O_3_ crucible shows the worst slag corrosion resistance.

(2) The addition of CA_6_ to the raw materials has a positive effect on improving the slag corrosion resistance of the Al_2_O_3_ crucible.

(3) The generation of high melting point CaAl_4_O_7_ is the critical factor for inhibiting the further corrosion of the CA_6_ and CA_6_-Al_2_O_3_ crucible. The CaAl_4_O_7_ was also detected in the Al_2_O_3_ crucible, but Al_2_O_3_ in the refractory reacts with CaO in the refining slag to produce CaAl_12_O_19_ firstly, and then the CaAl_12_O_19_ reacted with slag to form CaAl_4_O_7_. Therefore, the generation of CaAl_4_O_7_ in Al_2_O_3_ crucible is slower than that of CA_6_ crucible and the amount generated is relatively less, which results in a worse slag corrosion resistance of Al_2_O_3_ crucible compared to CA_6_ crucible and CA_6_-Al_2_O_3_ crucible.

(4) When the refining slag reacts with the crucible, the double pyramid module of CA_6_ will be destroyed quickly, resulting in the rapid collapse of the CA_6_ crystal structure to form the denser CaAl_4_O_7_, which is an essential reason for the excellent slag corrosion resistance. At the same time, it also avoids CA_6_ particles entering the molten steel to introduce exogenous inclusions, so CA_6_ has great application potential in ladle refining and clean steel smelting.

## Figures and Tables

**Figure 1 materials-15-06779-f001:**
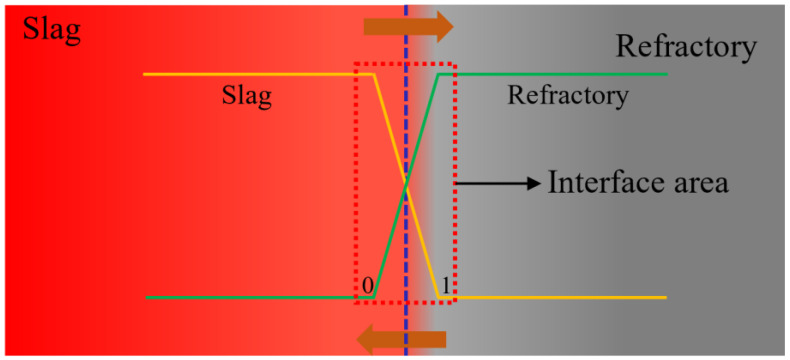
Thermodynamic calculation mechanism of the corrosion process.

**Figure 2 materials-15-06779-f002:**
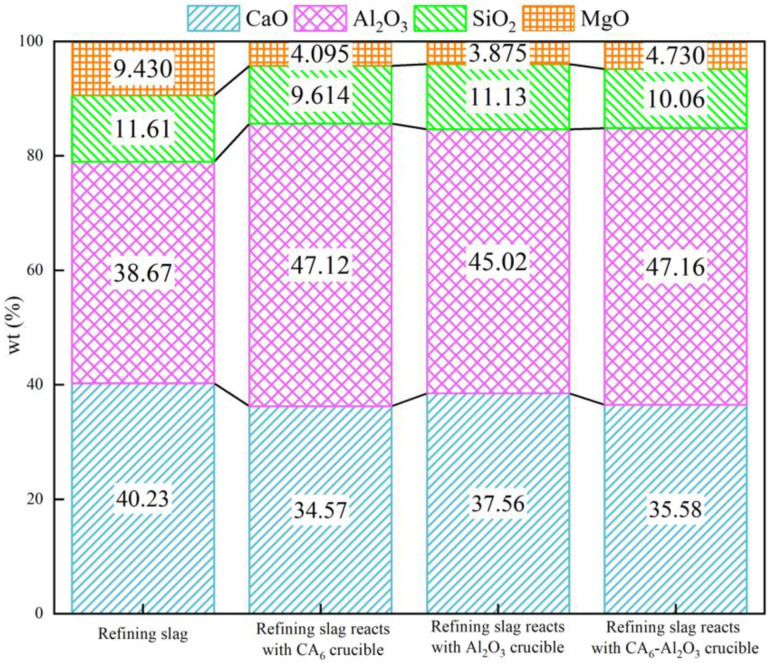
Composition variation of the refining slag.

**Figure 3 materials-15-06779-f003:**
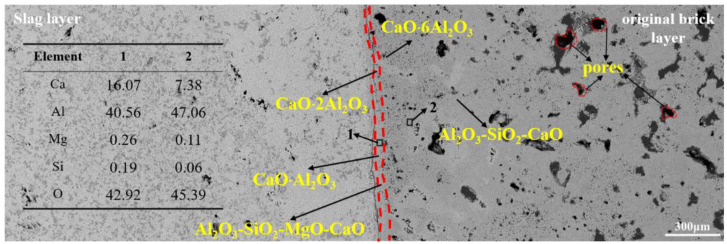
SEM image of CA_6_ crucible after corrosion.

**Figure 4 materials-15-06779-f004:**
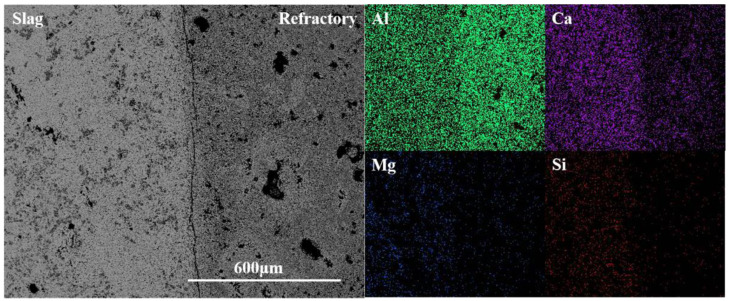
Element mapping of the slag-crucible interface of CA_6_ crucible.

**Figure 5 materials-15-06779-f005:**
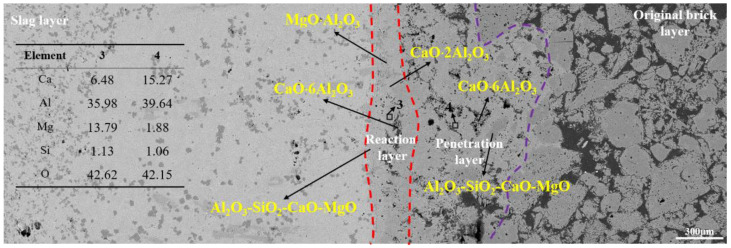
The SEM image of the Al_2_O_3_ crucible after corrosion by slag.

**Figure 6 materials-15-06779-f006:**
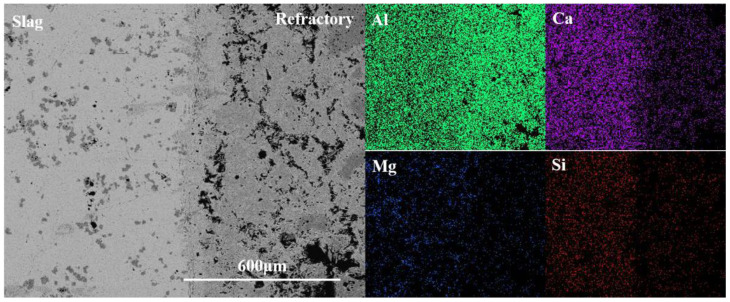
Element mapping of the slag-crucible interface of Al_2_O_3_ crucible.

**Figure 7 materials-15-06779-f007:**
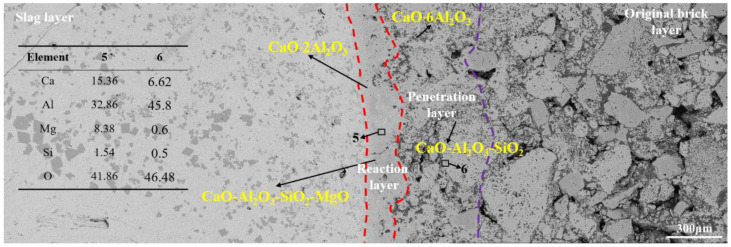
The results of the CA_6_-Al_2_O_3_ crucible after corrosion by slag.

**Figure 8 materials-15-06779-f008:**
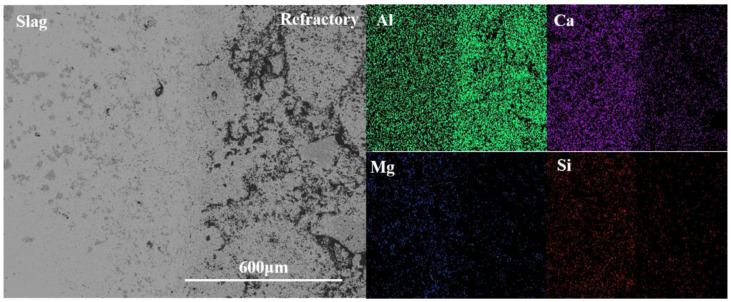
Element mapping of the slag-crucible interface of CA_6_-Al_2_O_3_ crucible.

**Figure 9 materials-15-06779-f009:**
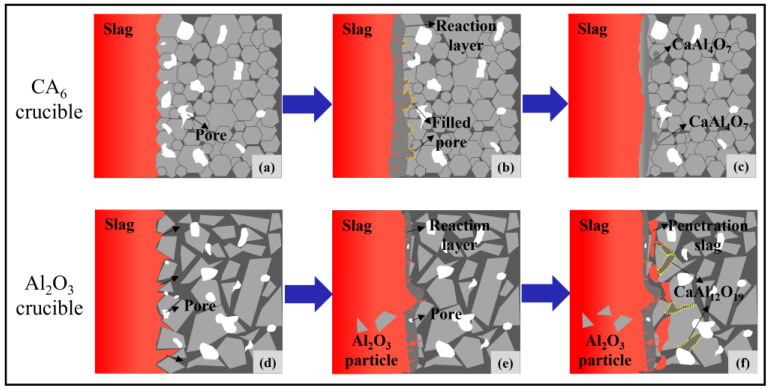
Reaction mechanism of the slag corrosion on the CA_6_ and Al_2_O_3_ crucible. (**a**–**e**): The corrosion process of the CA_6_ crucible; (**d**–**f**): The corrosion process of the Al_2_O_3_ crucible.

**Figure 10 materials-15-06779-f010:**
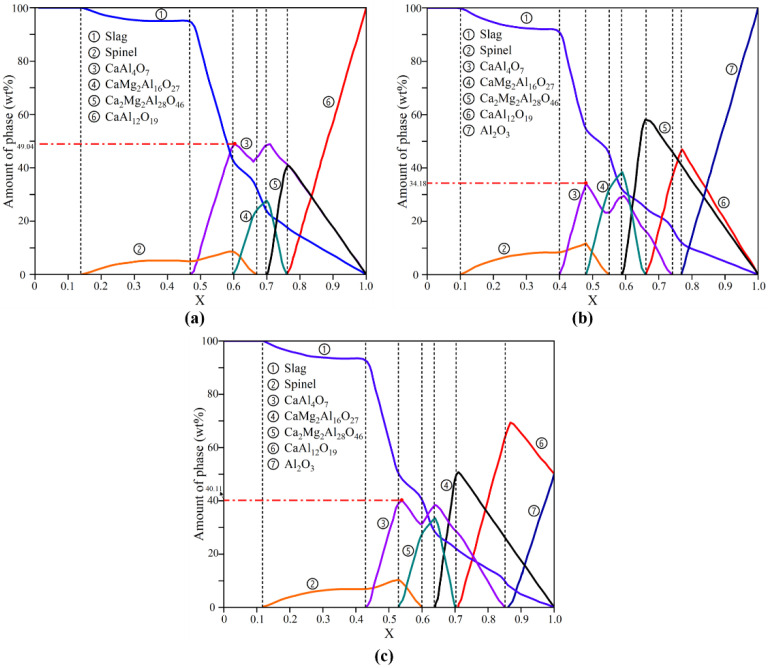
The thermodynamic simulation results of three crucibles: (**a**) CA_6_ crucible, (**b**) Al_2_O_3_ crucible, and (**c**) CA_6_-Al_2_O_3_ crucible.

**Figure 11 materials-15-06779-f011:**
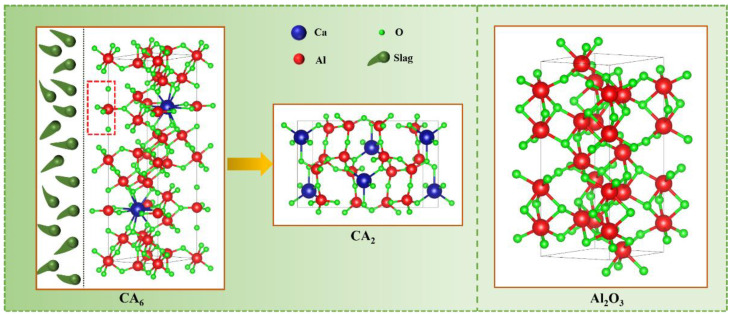
The crystal structure of CA_6_, CA_2,_ and Al_2_O_3_.

**Table 1 materials-15-06779-t001:** The properties of the crucibles.

Crucible	Bulk Density/(g/cm^3^)	Apparent Porosity/(%)
CA_6_	2.93	19.23
Al_2_O_3_	2.92	24.41
CA_6_-Al_2_O_3_	2.68	28.93

**Table 2 materials-15-06779-t002:** Chemical compositions of the refining slag (wt%).

	CaO	Al_2_O_3_	MgO	SiO_2_
Wt (%)	40	39	10	11

## Data Availability

Not applicable.

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
