# Peer review of "Reaction Mechanism of CA6, Al2O3 and CA6-Al2O3 Refractories with Refining Slag"

_materials, 2022, doi:10.3390/ma15196779_

Round 1

Reviewer 1 Report

The manuscript "Reaction Mechanism of CA6, Al2O3 and CaO-Al2O3 Refractories with Refining Slag" investigates and simulates the corrosion mechanism caused by refining slag in refractories of the Al2O3 system. The results obtained in this study indicate that CA6 refractory has the potential to be used in ladle refining and clean steel smelting due to the formation of a reaction layer at the slag-refractory interface (CaAl4O7) that prevents further slag infiltration.

The manuscript is well structured and can be accepted for publication after addressing the following points:

1) In section 2.1. "Preparation of crucibles and refining slag", erosion test is mentioned, which must be an error since the static crucible method is applied to evaluate the corrosion resistance of refractories. On the other hand, the dimensions of the cubic samples have to be expressed correctly.

2) In section 3.3. "Thermodynamic simulation of corrosion of crucibles by refining slag" as well as the "calculation mechanism and parameter setting", should be included in section 2 after the characterization as a complement to understand the corrosion process.

3) Considering the Al2O3 crucible, in lines 175-176 it is mentioned that the slag with Al2O3-CaO-SiO2-MgO phase is found even in the penetrating layer, which indicates that the reaction layer has little advantage on preventing the slag penetration. Does the presence of the MgO-Al2O3 phase have any influence on this behavior of the protective layer?

4) The equations of chemical reactions must be expressed correctly.

5) In lines 277 and 278, when 0.76<X<0.7 the CaMg2Al16O27 phase appears and increases with increasing X. This appears to be an incorrect statement, once Figure 10(a) is reviewed.

6) Lines 309 and 310 emphasize that the improvement in corrosion resistance is due to the formation of a phase with a high melting point CaAl4O7; this melting point should be included as a reference.

Reviewer 2 Report

This article describes the corrosion behavior of CA6 based refractory with refining slag (composed of oxides). The findings are interesting, however, I have the following minor comments to the authors. 

1. The abstract need to be reorganized such that it reflects the content of the paper and the main findings. 

2. There are fair amounts of typos and grammatical errors in the paper, thus language revision is needed. 

3. Page 2 line 77, the third dimension of the compressed dried powder is missing.

4. Adequate explanation of the thermodynamic simulation methodology should be added to the materials and methods section.

5. Page 3 line 108-109, what are the chanced that CaO and MgO formed reacted with other material rather than entered into the refractories? These materials are highly reactive under high temperature. 

Round 2

Reviewer 2 Report

The authors addressed all the comments and I think the paper can be published